# Assessing the Influence of the Sourcing Voltage on Polyaniline Composites for Stress Sensing Applications

**DOI:** 10.3390/polym12051164

**Published:** 2020-05-19

**Authors:** Andrés Felipe Cruz-Pacheco, Leonel Paredes-Madrid, Jahir Orozco, Jairo Alberto Gómez-Cuaspud, Carlos R. Batista-Rodríguez, Carlos Andrés Palacio Gómez

**Affiliations:** 1Facultad de Ciencias, Universidad Pedagógica y Tecnológica de Colombia, Tunja 150003, Colombia; andresfelipe.cruz@uptc.edu.co (A.F.C.-P.); jairo.gomez01@uptc.edu.co (J.A.G.-C.); 2Max Planck Tandem Group in Nanobioengineering, University of Antioquia. Complejo Ruta N, Calle 67 No. 52–20, Medellín 050010, Colombia; grupotandem.nanobioe@udea.edu.co; 3Faculty of Mechanic, Electronic and Biomedical Engineering, Universidad Antonio Nariño, Tunja 150002, Colombia; carlos.batista@uan.edu.co; 4Faculty of Sciences, Universidad Antonio Nariño, Tunja 150002, Colombia; carlospalacio@uan.edu.co

**Keywords:** polyaniline, polyethylene, piezoresistive sensor, pressure sensor, polymer composite

## Abstract

Polyaniline (PANI) has recently gained great attention due to its outstanding electrical properties and ease of processability; these characteristics make it ideal for the manufacturing of polymer blends. In this study, the processing and piezoresistive characterization of polymer composites resulting from the blend of PANI with ultra-high molecular weight polyethylene (UHMWPE) in different weight percentages (wt %) is reported. The PANI/UHMWPE composites were uniformly homogenized by mechanical mixing and the pellets were manufactured by compression molding. A total of four pellets were manufactured, with PANI percentages of 20, 25, 30 and 35 wt %. Fourier-transform infrared (FTIR) spectroscopy, thermogravimetric analysis (TGA), differential thermal analysis (DTA), scanning electron microscopy (SEM) and energy-dispersive X-ray spectroscopy (EDS) were used to confirm the effective distribution of PANI and UHMWPE particles in the pellets. A piezoresistive characterization was performed on the basis of compressive forces at different voltages; it was found that the error metrics of hysteresis and drift were influenced by the operating voltage. In general, larger voltages lowered the error metrics, but a reduction in sensor sensitivity came along with voltage increments. In an attempt to explain such a phenomenon, the authors developed a microscopic model for the piezoresistive response of PANI composites, aiming towards a broader usage of PANI composites in strain/stress sensing applications as an alternative to carbonaceous materials.

## 1. Introduction

Polyaniline (PANI) has been positioned as a relevant material in recent decades due to its outstanding electromechanical characteristics and attractive processing properties. Despite being discovered in the nineteenth century, the usage of PANI remained rather scarce until the 1980s when highly conductive PANI was synthesized for the first time [1,2]. The conductivity of PANI can be trimmed through grafting or irradiation [3,4] while exhibiting environmental stability to external agents [5,6]. Besides this, PANI is a low-cost polymer with multiple ways of preparation [7]. PANI is commonly synthesized by polymerization of aniline in an aqueous acidic medium, prioritizing the formation of the most conductive form of PANI, the emeraldine salt (ES). The properties of PANI-ES depend on the type of dopant and the level of protonation, the size of the dopant ion, and the preparation conditions (temperature, monomer/oxidant ratio, etc.). In this sense, organic sulfonic acids have been the most widely used dopants to produce more stable and conductive polyaniline [8,9]. For these reasons, the inclusion of PANI and PANI composites has been thoroughly investigated as an alternative material in multiple applications, such as electromagnetic shielding [10], antennas manufacturing [11,12], supercapacitors [13] and as chemical sensors [14,15].

Regarding sensing applications, PANI composites have been successfully tested for measuring the concentration of pollutant gases, such as ammonia [16], carbon monoxide [17] and hydrogen sulfide [18]. Likewise, mechanical variables can be also measured from PANI composites. Valentová et al. investigated the piezoresistive response of PANI pellets under compressive loading [19]. Tensile stress [20] and bending [21] have been measured using polymer blends of PANI and styrene–butadiene–styrene (SBS). The usage of ternary polymer composites has been also investigated for strain sensing [22].

Nonetheless, the usage of PANI composites for mechanical measurements does not dominate in nowadays applications; but instead, polymer composites comprising CNT, carbon black (CB), or metallic particles are preferably chosen over PANI [23]. To understand this preference, one must define the gauge factor (*GF*) as the relative variation in the electrical resistance (Δ*R*) to the relative variation in dimensional length (Δ*l*), i.e., *GF* = Δ*R*/Δ*l*. In general, materials exhibiting large *GF*s are desirable for sensor development because a small change in the measured parameter (length or stress) causes a significant variation in the electrical resistance. Unfortunately, the *GF* of PANI composites has been experimentally measured to be around 1 or 2 at most [20,21,24,25,26], whereas CB/CNT polymer composites exhibit *GF*s typically around 4 [27], 10 [28], or even higher [29,30,31]. Fortunately, recent investigations have demonstrated that with proper doping, the *GF* of PANI composites can be increased up to 10 [32,33], 50 [34] or even 100, as demonstrated by Falletta et al. [35]. Template polymerization of polyaniline has been also investigated; Sezen-Edmonds et al. demonstrated that the molecular weight of the template polymer plays a major role in the piezoresistive response of PANI [26].

Besides the advantages of polyaniline, current research interest on PANI has arisen from the potential hazard of CNT/CB polymer composites. As mentioned by Costa et al. [20], carbonaceous materials (CNT, CB and graphene nanoplatelets) may be toxic [36,37], and therefore, wearable devices that incorporate carbonaceous materials must be treated with special consideration to avoid potential health disorders. Wearable devices are smart electronic apparatus aimed to be installed or worn on the human body or clothing [25,38,39,40,41].

Similarly, the investigation of support matrix materials is also an active research trend, especially in those applications involving piezoresistive sensing. Recently, ultra-high molecular weight polyethylene (UHMWPE) has been used as a polymeric matrix since it is a high-performance thermoplastic polymer [42,43]. UHMWPE is commonly used for the manufacture of implants due to its excellent mechanical, tribological and chemical properties that, combined with a high degree of biocompatibility and low-cost, make it attractive for its use in biomedical applications [44,45]. Particularly, few reports using UHMWPE have focused on analyzing the electromechanical properties under stress, perhaps due to the extensive molecule chain that generates some limitations in terms of viscosity and processing methods [42]. These inconveniences could be solved by reinforcing the UHMWPE particles with inexpensive polymeric elements, and by proposing new processing routes to generate integrated structures during the molding stages [46]. In this way, the development of UHMWPE—and intrinsic semiconductor polymer-based composites—could be of great benefit for the development of piezoresistive materials with superior mechanical and electrical properties. These properties will be beneficial in industrial and biomedical applications [47,48].

Aiming to promote the usage of PANI composites in wearable devices, this study is focused on investigating the piezoresistive response of PANI–polyethylene composites under compressive loading (henceforth referred to as PANI/UHMWPE composites). Typically, the piezoresistive response of PANI composites is studied under conditions of compressive stress [19,24]. Nonetheless, a different methodology was followed in this study with two relevant aspects: first, PANI/UHMWPE pellets have been prepared and tested at multiple sourcing voltages (*V*_s_). This was done for assessing the influence of *V*_s_ over the performance metrics of hysteresis and drift errors, being a novel aspect of the present investigation since previous studies were performed under conditions of constant *V*_s_ [20,21,24,25,26,32,33,35]. Second, a physical model for the piezoresistive response of PANI/UHMWPE pellets has been derived and tested. The model is based on previous studies from Mikrajuddin et al. [49] and Shi et al. [50] for the constriction resistance of contacts. The methodology and results from this research aim to develop stress sensors based on PANI/UHMWPE blends, which may ultimately replace polymer sensors that incorporate toxic carbonaceous materials such as CB and CNT. Likewise, the performance analyses at multiple voltages are intended to find the optimal *V_s_* that minimizes the error metrics during stress measuring.

## 2. Theoretical Models for the Temperature- and Voltage-Dependent Response of Polyaniline

Multiple authors have extensively investigated the conduction mechanism of polyaniline since the 1970s; Sheng [51], and Mott and Davis [52] gave the first steps in such a direction, and more recently, Lin et al. [53], and Kang and Snyder [54] further improved the conduction models of PANI and PANI composites. The operating temperature (*T*) and the sourcing voltage, *V*_s_, do play a significant role in the conductivity of PANI. Given conditions of constant stress and temperature, changes in *V*_s_ cause a non-linear variation in the PANI current (*I*) [20,21,35]. Similarly, given conditions of constant stress and *V*_s_, the resistance of polyaniline changes in a non-linear fashion with temperature increments [53,55,56]. However, to the best of the authors’ knowledge, no author has studied how the sourcing voltage modifies the piezoresistive response of PANI composites at the microscopic level. Costa et al. and Falletta et al. have proposed a model for the piezoresistive response of PANI [20,35], but such a model is a macroscopic approach that is based on a sensor’s dimensional change and Poisson ratio. In the next section, a microscopic model is proposed for the piezoresistive response of polyaniline.

### 2.1. A review of the Mott’s Model for the Electron Conduction in Granular Metals

Polyaniline conduction is originated from electron hopping among neighboring conducting grains, which are separated from each other by insulating grains. The conducting grains are protonated (emeraldine salt), whereas the insulating ones are unprotonated (emeraldine base), as illustrated by Lin et al. [53]. The average inter-grain separation between conducting monomers is represented by *s*, which is typically in the range of a few nanometers [53].

According to the Mott’s model [52], the electric-field-dependent conductance of PANI, *G*(*E*), can be found from *G*(*E*) = *G*_0_exp(*esE*/*k*_B_*T*), where *e* is the electron charge, *s* is the inter-grain separation, *E* is the applied electric field, *k_B_* is the Boltzmann’s constant, *T* is temperature, and *G*_0_ is the conductance measured at the lowest admissible *V_s_*. Such expression holds when *E* is lower than 10^3^ Vcm^−1^ (low-voltage regime), which is the case of most strain/stress sensors. Typically, the longitudinal dimension (*d*) of a strain/stress sensor is within the fraction of a centimeter (0.02 cm in the thinnest case), and therefore, voltages up to 20 V can be applied without violating our premise, where *E* = *V*_s_/*d*. On the other hand, the inter-grain separation can be related to the applied stress (*σ*) from the formula
(1)$$s=s_{0}\left(1-\varepsilon \right)=s_{0}\left(1-\sigma /M\right)$$
where *ε* is the resulting strain, *s*_0_ is the uncompressed inter-grain separation, and *M* is the compressive modulus of polyaniline. From a theoretical standpoint, incremental stresses cause a reduction in the inter-grain separation, which ultimately reduce *G*(*E*) according to the Mott’s model; this may seem contradictory at a glance, but the hopping mechanism is a two-fold phenomenon that embraces the inter-grain separation and the Coulomb charging energy that also depends upon *s*. Further details of the hopping mechanism can be found at [57,58]. Conversely, when a PANI composite is elongated, an increment in the sample’s conductance is theoretically expected.

For a better understanding of the Mott’s model, Figure 1 shows a simplified version of *G*(*E*) such that the constant *G*_0_ has been taken out to study how *s* and *V_s_* influence *G*(*V*_s_). The substitution *E* = *V*_s_/*d* has been also done as the following:(2)$$G\left(V_{s}\right)=\left(G\left(E\right)-G_{0}\right)/G_{0}=exp(esV_{s}/dk_{B}T)-1$$

The plot in Figure 1 shows *G*(*V*_s_) for multiple inter-grain separations at *T* = 300 K and *d* = 0.02 cm. Note that increasing *s* or *V_s_* has the same effect on *G*(*V_s_*)—they are both multiplied in Equation (2)—but changes in *G*(*V*_s_) are rather small, even if *s* is doubled or tripled; this is so because the argument of the exponential is quite small (around 10^−2^). Nonetheless, strain sensors manufactured from PANI composites have demonstrated that small elongations can produce significantly higher variations than those predicted from Equation (2) [32,33,34,35].

Besides this, the PANI strain sensors manufactured by Falletta et al. [35] and Costa et al. [20] showed that elongations of PANI composites produce a decrement in the sensor’s conductance. On the other hand, experimental observations from Valentová et al. demonstrated that PANI compression causes an increment in composite conductance [19]. In brief, experimental observations contradict the theoretical predictions of Mott’s model plotted in Figure 1.

At this point, it must be clarified that the Mott’s model has been thoroughly tested, so its validity is undoubted [53]; later works from Kang and Snyder [54] have further improved the original Mott’s formulation, but the underlying basis of the Mott’s model has remained unchanged to date. The fact that experimental observations do not agree with Mott’s model is a clear indication that additional phenomena may be occurring at a microscopic level that we are currently ignoring.

### 2.2. Proposal of a Piezoresistive Model for PANI/UHMWPE Composites Based Upon Changes of the Constriction Resistance

Special attention must be placed on the experimental results if a valid model is developed for the piezoresistive PANI composites. In this regard, the work from Valentová et al. reported that the sensor’s conductance saturated upon application of large stresses (over 200 MPa); this is a clue that indicates a plastic deformation in the material microstructure. Based on this observation, the following microscopic model is proposed for the piezoresistive variation of PANI composites
(3)$$R\left(\sigma \right)=R_{0}+R_{1}/\left(\sigma +\sigma_{0}\right)$$
where *R*(*σ*) is the total resistance resulting from the sum of two elements: the nanocomposite resistance, *R_0_*, and the contact resistance, *R_1_*/(*σ* + *σ*_0_), which is a stress-dependent magnitude. The term *R_0_* is originated from electron hopping among neighboring conductive grains following the Mott’s model (*G*_0_=1/*R*_0_ as defined in Equation(2)), whereas *R_1_*/(*σ* + *σ*_0_) is originated from the plastic deformation occurring between the conductive polymer grains and the metallic electrodes. This formulation has been theoretically modeled and experimentally verified by Mikrajuddin et al. [49] and Shi et al. [50]. The contact resistance depends upon the application of stress in an inverse proportionally fashion, but with an effective offset (*σ*_0_); this offset can be understood as the pre-compression applied to the nanocomposite during assembly. Finally, the factor *R*_1_ depends upon the physical dimensions of the constriction paths. Note from Equation (3) that the nanocomposite resistance, *R*_0_, has been written as a stress-independent magnitude; this is a logical consequence of the simulations plotted in Figure 1. The minus one power in the contact resistance term, (*σ* + *σ*_0_)^−1^, is typical of plastic deformations, which do occur in polymers. Other power factors, such as −1/3 or −2/3 are only observable in elastic deformations [49,50].

In Section 4, experimental data are fitted to Equation (3) to assess its validity as a piezoresistive microscopic model for PANI composites. It must be stated that Equation (3) was given in terms of resistance to be consistent with the original formulation from Mikrajuddin et al. [49] and Shi et al. [50]. However, conductance and resistance magnitudes are indistinctly used in this paper, depending on the context, where *R* = 1/*G*.

## 3. Materials and Methods

### 3.1. Materials

Commercial polyaniline (emeraldine salt) powder doped with organic sulfonic acids used in this study was purchased from Sigma-Aldrich (San Louis, MO, USA, product no: 428329), with a particle size of 3–100 µm and an average molecular weight >15,000 g mol^−1^. UHMWPE also from Sigma-Aldrich (San Louis, MO, USA, product no: 434272) had a particle size of 40–48 µm, with a density of 0.94 g mL^−1^ at 25 °C. The ethanol used for the solution mixing was of analytical grade.

### 3.2. Composites Preparation

UHMWPE composites with different weight percentages (wt %) of polyaniline (20, 25, 30 and 35 wt %) were fabricated by mixing the powders in solution, followed by compression molding technique [43,59]. Costa et al. [20] and Andreatta et al. [60] experimentally determined the percolation threshold to be around 10 wt % for PANI composites, and consequently, the prepared pellets in our study were operating under the percolation regime. Figure 2 shows the general scheme of the sample preparation process.

Initially, PANI was dispersed in ethanol and ultrasonicated at 40 kHz for 30 min. UHMWPE powder was dispersed separately in ethanol using magnetic stirring [59]. Then, UHMWPE/ethanol dispersion was mixed with the PANI/ethanol dispersion and ultrasonication continued for 20 min. The final suspension was placed in a fume hood at room temperature for 24 h to evaporate the solvent completely. The PANI/UHMWPE powder after the evaporation mixture was further milled in an agate mortar for 2 h. Ethanol did not affect the conformation of PANI chains doped with organic sulfonic acids and preserved its morphology and electrical conduction properties [61]. Finally, the macerated and mixed powders of different PANI wt % were compressed and molded in a hydraulic press at 10 MPa for 10 min, followed by a sintering process at 170 °C for 10 min; this was done to obtain rigid pellets of 9 mm diameter and 2 mm thickness, which were finally cooled naturally at room temperature [42,62]. The electrical contact between the pellet and the electrodes was improved by coating the bottom and top surfaces with silver ink. The silver-coated composites were sandwiched between copper electrodes for later electromechanical tests.

### 3.3. Characterization of Composites

The infrared spectra were measured in a Thermo Scientific Nicolet iS50 spectrometer (Madison, WI, USA), equipped with a diamond crystal by the attenuated total reflection (ATR) technique. Each sample was placed on the ATR accessory diamond crystal and the measurements were made using 32 scans with a resolution of 4 cm^−1^. Before the analysis, the background was measured under the same conditions with a clean empty cell. The TGA and DTA were developed in a TA instrument simultaneous thermal analyzer, model SDT Q600 (Champaign, IL, USA). Samples were analyzed under a nitrogen atmosphere with 20 mL min^−1^ flow, from 25 to 700 °C, at 5 °C min^−1^. The morphological characterization of the PANI/UHMWPE composites was analyzed by scanning electron microscopy (SEM) with a gold coating on a JEOL-JSM 6490LV microscope (Tokyo, Japan), with an 15 KV acceleration voltage and operating a secondary electron scattering under high vacuum conditions. Energy-dispersive X-ray spectroscopy (EDS) microanalysis was performed on a Phenom ProX EDS System (Eindhoven, The Netherlands), equipped with a silicon detector for light elements and resolution of 132 eV.

The experimental set-up comprised a tailored test bench for applying the stress profiles and an amplifier in inverting configuration to readout the resistance of the pellet. The test bench embraced a linear motor, a load cell, and a spring connected in series with the motor shaft to provide mechanical compliance (Tunja, Colombia). Additional details of the test bench were already provided in previous authors’ work [63]. The PANI/UHMWPE pellets had a diameter of 9 mm. The maximum applied force was 68.6 N; thus, the full stress matched for 1.08 MPa.

The driving circuit for measuring the resistance of the pellet comprised the LF353 amplifier connected in an inverting configuration. This configuration ensured that the desired *V_s_* was always applied to the specimen under test [64]. Unless otherwise noted, the test voltages of 1.25, 2.5, 5 and 7.5 V were applied to the PANI/UHMWPE pellets. Table 1 shows the pellet’s conductance (*G*) measured at *V*_s_ = 1 V.

Note from Table 1 that conductance does not necessarily increase with larger PANI content; this is caused by a wide range of factors, e.g., random distribution of the PANI percolation network in the composite, and roughness between the PANI/UHMWPE pellet and the electrodes causing contact area to be stress-dependent, thus influencing the conductance at null stress.

## 4. Results and Discussion

### 4.1. Structural, Thermal and Morphological Characterization

FTIR spectroscopy, shown in Figure 3, was used to investigate the successful process of mixing PANI with UHMWPE in the formation of composite materials. The FTIR spectrum of the UHMWPE shows two characteristic bands of the asymmetric and symmetrical stretch vibration modes of the bond –CH_2_ at 2914 cm^−1^ and at 2847 cm^−1^, respectively. The band at 1471 cm^−1^ corresponds to the bending mode of the C–H and the one at 717 cm^−1^ represents the rocking mode vibration of the C–H [44,65]. Likewise, the FTIR spectrum of the PANI and the composites show the characteristic bands of the conducting emeraldine salt phase in the region between 1560 and 1530 cm^−1^ attributed to the stretching vibration of N=Q=N bonds of quinoid diamine ring (Q) and the C–C aromatic ring stretching vibration of the benzenoid diamine in 1460 cm^−1^ [66]. The band at 1295 cm^−1^ is attributed to the C–N stretch bond of the aromatic amines of the PANI. The 1100 cm^−1^ band corresponds to the in-plane bending vibration of a benzene ring and is related to the presence of charge carriers along the polymer chain. The peak at 825 cm^−1^ is due to the C–H out-of-plane bending vibration of the substituted benzene at position 1,4 [48]. The small peak at 3225 cm^−1^ is due to the N–H stretching vibration. The bands at 1040–1006 cm^−1^ and 567 cm^−1^ are related to S=O and S–O stretching vibrations, respectively, thus confirming the doping of the conductive phase of PANI emeraldine salt with organic sulfonic acids (R-SO_3_) [67,68]. In this context, the FTIR spectra of the PANI/UHMWPE composites show a decrease in the characteristic bands of the UHMWPE proportional to the increase in the proportion of PANI, confirming the successful formation of the composite materials.

The TGA and DTA curves of the composite with 20, 25, 30 and 35 wt % PANI content are presented in Figure 4a,b. The TGA and DTA were used to confirm the different wt % fraction of polyaniline in the composites and to evaluate the thermal stability of the compressed and heat-treated materials. The thermal decomposition of UHMWPE occurs in a single stage between 300 and 500 °C, while polyaniline exhibits two stages of weight loss. The first stage of loss, before 150 °C, corresponds to the evaporation of doping molecules present in the emeraldine form of polyaniline. The second weight loss after 200 °C is attributed to the decomposition of oligomers and the degradation of the polyaniline macromolecular chain [69]. The thermal decomposition of PANI/UHMWPE composites can be summarized as a three-step process. Thus, the TGA curves of the composites show an increase in weight loss as the proportion of polyaniline changes from 20 to 35 wt %, which is related to the partial and complete degradation of PANI and UHMWPE in the composites [48].

Similarly, the DTA curve shows the heat flow changes in each component of the PANI/UHMWPE composite. DT analysis of UHMWPE shows an endothermic peak at 144 °C, corresponding to the melting point for UHMWPE. The second exothermic peak at 235 °C is related to oxidation of the polyethylene and the beginning of its degradation that continues until 465 °C. At this point, a slight exothermic peak at 468 °C is related to the total combustion of char residue [70]. Thus, the melting point signal of polyethylene serves as a reference to study the increase in the proportion of PANI at the polyethylene interfaces. In this context, when the amount of PANI increased, the intensity of the characteristic peaks in the polyethylene decreased.

To highlight, crystallization and semi-crystallization induced by deformation at temperatures above those of the melting point of UHMWPE are beneficial to decreasing the percolation limit and consequently improving the characteristic electrical conductivity of the composite [71]. This phenomenon is produced by the location of conductive particles at the interfaces of the substrate particles. It generates a deformation hardening, like the behavior of natural rubber, that provides flexibility to the pellets, ideal for applications as stress-sensors [42,72,73,74]. This effect is also associated with a higher degree of CNT–polymer intermingling in the interfacial regions from the lower polymer viscosity at higher temperatures [62]. Semi-crystallization induced by deformation grants’ stability to the composite as the PANI was embedded in the partially molten UHMWPE. This effect was verified in the DTA curve of the different composites. The signal intensity, for example, of the UHMWPE melting point peak, decreased when the amount of PANI increased by checking the effective location of PANI at the interfaces of the UHMWPE particles.

Figure 5 shows the SEM images of the composites with different wt % of PANI. The micrographs show the homogeneous dispersion of the PANI particles around the UHMWPE particles. The SEM results corroborated control in the addition of the two polymers during pellets assembly in agreement with the FTIR, TGA and DTA techniques. In the micrographs, the PANI particles show a bright contrast due to their high electron density. Conversely, the UHMWPE insulating particles are rather dark. Thus, by increasing the amount of PANI from 20 to 35 wt %, the UHMWPE particle surface is comparatively more coated by the conductive polymer, and therefore the electrical conductivity is enhanced, which is ultimately required in piezoresistive sensors.

Besides, the SEM images show the wear of the UHMWPE particles in the form of fibers, which is associated with the partial melting processes that the material suffered in the sintering stage, but without losing its mechanical properties, as discussed in previous work [62,72]. As mentioned in the DT analysis, the semi-crystallization induced by the deformation of polyethylene at high temperatures and with PANI particles at the interfaces allows the manufacture of highly homogeneous and flexible conductive pellets.

The EDS spectra of the PANI/UHMWPE composites shown in Figure 6 contain peaks associated with C, O, N and S confirming the emeraldine salt form of PANI doped with organic sulfonic acids R-SO_3_. Atomic percentage data for each element are displayed on the EDS spectra. The data indicate increases in the atomic percentages of O, N and S and decreases in C, according to the increase in wt % of PANI doped with sulfonic groups in the UHMWPE matrix. These data agree with the results of FTIR, TGA, DTA and SEM analysis.

### 4.2. Electrical and Piezoresistive Characterization

The piezoresistive characterization of the PANI/UHMWPE pellets was carried out based on two tests: a voltage sweep to determine the current-voltage characteristic (*I*-*V*), and compression tests at different voltages. The sweep was done at null stress, starting at *V*_s_ = 0 V up to 7.5 V with a pace of 0.1 V; see Figure 7. In general, the PANI/UHMWPE composites exhibited somewhat high linearity, just as reported in previous studies [20,21,35,53]. However, a closer look at the *I*-*V* data showed some nonlinearities occurring in the nanocomposite, especially for voltages under 1 V. The main consequence of the *I*-*V* nonlinearity is that Equation (3) requires individual fitting for each *V*_s_, which is later discussed.

Figure 8 reports the piezoresistive response of the PANI/UHMWPE pellets measured at the test voltages. The trend lines obtained from data fitting are also shown in Figure 8, where Equation (3) was used as the fit model. All the fits were successful with a determination coefficient (*R^2^*) greater than 0.98. Considering that the pellets’ resistance is voltage-dependent, individual fits were done at each *V*_s_; the parameters resulting from the fits are summarized in Table 2. Note that for a given pellet, *R*_0_ and *σ*_0_ remain rather constant for each fit; this is a clear indication that such magnitudes are voltage-independent. It is not a surprising result given the fact that Figure 1 predicted a little variation in the nanocomposite resistance, *R*_0_. Likewise, nanocomposite pre-compression, *σ*_0_, is fundamentally a mechanical variable, thus unaffected by *V*_s_.

Regarding *R*_1_, it is not convenient to study it directly from Figure 8 and/or Table 2; this is so because such plots embrace two factors simultaneously: (i) the *R_1_* dependency on *V*_s_, and (ii) the random variations occurring at the interface between the metal electrodes and the polymer. The latter factor originates from the fact that each *R*(KΩ) vs. *σ*(MPa) dataset was taken at different trials, thus unavoidably some conductions paths were disconnected among subsequent loadings (and others reconnected), i.e., the polymer surface is rough at the microscopic scale, just as previously shown in the SEM micrographs of Figure 5.

Better setup for studying *R_1_* can be obtained by following these steps, given the data from Figure 7: (i) compute a reference resistance (*R*_ref_) using the single datum at *V*_s_ = 0.1 V. (ii) calculate resistance values for voltages higher than 0.1 V; these data are henceforth designated as *R*(*V*). (iii) Given *R*(*V*), the quotient *R*(*V*)/*R*_ref_ is then obtained and plotted in Figure 9. Given the fact that the whole *I*-*V* data points were taken in a single trail for each pellet, it is possible to set outplay the random disconnections/reconnections that occur in the interface between the polymer and the metal electrodes.

It is clear from Figure 9 that *R*(*σ*) is notably reduced for incremental *V*_s_, where *R*(*σ*) has been previously defined in Equation (3). Once again, it must be highlighted that this voltage-dependent behavior is only attributable to *R*_1_ itself and not to *R*_0_, i.e., the Mott’s model predicts little variation in the nanocomposite resistance, *R*_0_, see the plot in Figure 1. Unfortunately, we can only hypothesize about possible reasons for this observation as further tests are required for confirmation. In the works from Mikrajuddin et al. [49] and Shi et al. [50], the parameter *R*_1_ is defined as a function of the physical dimension and shape of the constriction paths. In our study case, polymer shape is unaffected by *V*_s_, but larger voltages enhance the activation of conductive grains, thus reducing the total resistance, *R*(*σ*).

An important distinction must be presented here. Electron hopping occurring between the polymer conductive grains and the metal electrodes is different from that occurring among conductive grains in the polymer; this is so because the former does not require a free hole to initiate hopping whereas the latter does, i.e., metallic electrodes can be assumed as an electron sink or reservoir depending on the sign of current flow. The practical consequence of this observation is that electron hopping between the electrodes and the conductive grains seems to be dramatically facilitated by incremental *V*_s_.

Finally, in the studies from Mikrajuddin et al. [49] and Shi et al. [50] the sourcing voltage is not a parameter because they were dealing with metals, which are isotropic materials. Conversely, the piezoresistivity of polyaniline is more similar to that of anisotropic materials that exhibit voltage dependency, such as CNT [75].

### 4.3. Assessment of the Piezoresistive Response at Multiple Voltages

Each specimen was loaded to 0 MPa and then to 1.08 MPa to assess the piezoresistive response of the PANI/UHMWPE pellets; this procedure was repeated at the aforementioned test voltages. With these data, the variation of conductance (Δ*G*) was computed using the following formula
(4)$$\Delta G=\left(G_{S}-G_{U}\right)/G_{U}$$
where GU stands for the uncompressed pellets’ resistance, and GS stands for pellets’ resistance at full stress. The results are presented in Figure 10.

It can be stated that lower Δ*G* were measured for incremental voltages regardless of polyaniline concentration, with a possible exception occurring for the specimen with 35 wt %. In practice, this implies that the sensor is less sensitive to force changes when operating at high *V*_s_.

### 4.4. Assessment of Hysteresis Error at Multiple Voltages

It is possible to define the hysteresis error (*HE*) from the formula
(5)$$HE=100\%\times \left(G_{u}-G_{l}\right)/G_{nom}$$
where Gu and Gl stand for the conductance measured during the unloading and loading phases at half nominal load (0.54 MPa), respectively. Gnom stands for the nominal conductance measured at the nominal stress of 1.08 MPa.

All the PANI/UHMWPE pellets were evaluated during a loading-unloading cycle to assess the influence of *V*_s_ over *HE*, and then, the *HE* was computed according to Equation (5). Figure 11 summarizes the results of this test. For voltages below 5 V, a clear trend was not observed in the hysteresis error, but for voltages over 5 V, the *HE* showed some reduction amid incremental voltages.

### 4.5. Assessment of Drift Error at Multiple Voltages

The drift error is defined as the relative change in conductance when a constant load is applied to a piezoresistive material. Given *G*_0_ as the instantaneous conductance of a pellet measured immediately upon the application of stress, the drift error (*DE*) can be calculated with the following expression
(6)$$DE\left(t\right)=100\%\times \left(G\left(t\right)-G_{0}\right)/G_{0}$$
where *G*(*t*) is defined as the time-varying conductance of the pellet.

The change in conductance is originated from the viscoelastic behavior of polymers that causes creep among neighboring conductive particles [76,77]. However, in our study case, it is expected that the major drift contribution is coming from creep occurring in the polymer-metal electrode interface. It is a reasonable hypothesis given the fact that *R*_1_/*σ*_0_ is considerably larger than *R*_0_; see Table 2. Besides, the Motts’ model predicts negligible variation in 1/*R*_0_ when *s* is changed; see Figure 1 and Equation (2).

To present the models that relate the inter-grain separation with time is out of the scope of this article; such models are known in the literature as rheological models and can be found in detail in Mainardi and Spada’s work [78]. In this manuscript, only experimental results of drift error are presented for the PANI/UHMWPE pellets. For this purpose, all the PANI/UHMWPE pellets were loaded to the constant stress of 1.08 MPa for one hour; the test was repeated at each test voltage. Figure 12a shows *DE*(t) for the 25 wt % pellet at multiple V_s_. Finally, Figure 12b summarizes the drift error measured at *t* = 3600 s for all the PANI/UHMWPE pellets. 

Figure 12a,b shows a clear sourcing voltage-drift error trend. Regardless of PANI content, incremental voltages reduce the drift error. This result is consistent with previous drift errors reported in polymer blends with CB [63].

### 4.6. Discussion and Relevance of Results

The development of polyaniline nanocomposites for strain/stress sensing is currently an active research trend. Given the multiple preparation methods and doping possibilities, it is common to find contradictory results in the specialized literature, e.g., some authors have reported negative GF in PANI strain sensors [79], whereas others have reported positive GF [33]. Sezen-Edmonds et al. made a step forward towards finding the basis of such a controversy; they found that the GF of PANI sensors is dramatically influenced by the molecular weight of the dopant [26]. Likewise, the proposal and validation of Equation (3) is a contribution to the same trend.

The piezoresistive model from Equation (3) explains—to some extent—the inconsistencies between the Mott’s model prediction and the experimental results (not only our results but also third authors’ results [32,33,34,35]). However, two questions remain unanswered: (i) why do some PANI composites exhibit positive GF and other negative GF? (ii) Why is *R_1_* so heavily dependent on *V*_s_ (just as reported in Figure 9)? Question (i) was partially answered by Sezen-Edmonds et al. [26] on a phenomenological basis; they provided explanations and arguments that support their observations, but a physical model was not presented. Concerning question (ii), the theory behind constriction resistance has been developed for metals—which are isotropic materials—but when dealing with granular metals such as polyaniline, further theoretical calculations are required.

The experimental results from Figure 10 through Figure 12 demonstrated that *V*_s_ plays an important role in sensors performance—especially in the drift error—but a theory of the constriction resistance is required for granular metals. This will let us understand why *R*_1_ is heavily influenced by *V*_s_ just as experimentally measured in Figure 9, to get a comprehensive understanding of these phenomena. The derivation of such a model is left as a future task for the authors.

The raw *I*-*V* data for plotting Figure 7 and Figure 9 are available at [80]. The whole set of conductance data as a function of stress and voltage is available at [81]; these data were employed for plotting Figure 8, Figure 10 and Figure 11. The time-sampled experimental data for drift assessment is available at [82]; these data were employed for plotting Figure 12.

## 5. Conclusions

Evenly dispersed PANI/UHMWPE composites were obtained by ultrasonication and mechanical milling. The finely mixed polymer powders were compressed and heat-treated to obtain piezoresistive pellets. The homogeneous distribution of PANI particles in the UHMWPE polymer matrix was characterized through FTIR, TGA and SEM analysis.

A microscopic model for the piezoresistive response of PANI composites was proposed and validated. The model embraced two separate contributions: (i) the nanocomposite resistance resulting from electron hopping in the PANI network, and (ii) the contact resistance occurring in the interface between the nanocomposite and the electrodes. The former contribution follows the Mott’s model for granular metals, whereas the latter is based on the constriction resistance of materials. It was experimentally demonstrated that the authors’ proposed model is a valid fit for the piezoresistive response of the PANI/UHMWPE composites.

The drift and hysteresis errors of the PANI/UHMWPE composites were assessed at different sourcing voltages (*V*_s_). It was found that *V*_s_ plays an important role in the drift error, where higher *V*_s_ resulted in lower drift error for all the nanocomposites. For the case of the hysteresis error, a clear trend was not observed regarding *V*_s_ changes; this was probably originated from the experimental setup that involved multiple loading/unloading cycles at different *V*_s_, and consequently, the disconnection and reconnection of conductions paths occurred between subsequent loadings.

The pellet with 25 wt % PANI content outperformed the other samples in the hysteresis and drift tests, observed regardless of the *V_s_* magnitude. Based on this observation, this weight ratio showed the optimal behavior in our study. Regarding *V*_s_, larger voltages lowered the error metrics but also lowered the conductance variation, and therefore an optimal *V*_s_ depends upon the specific requirements of the final application.

## Figures and Tables

**Figure 1 polymers-12-01164-f001:**
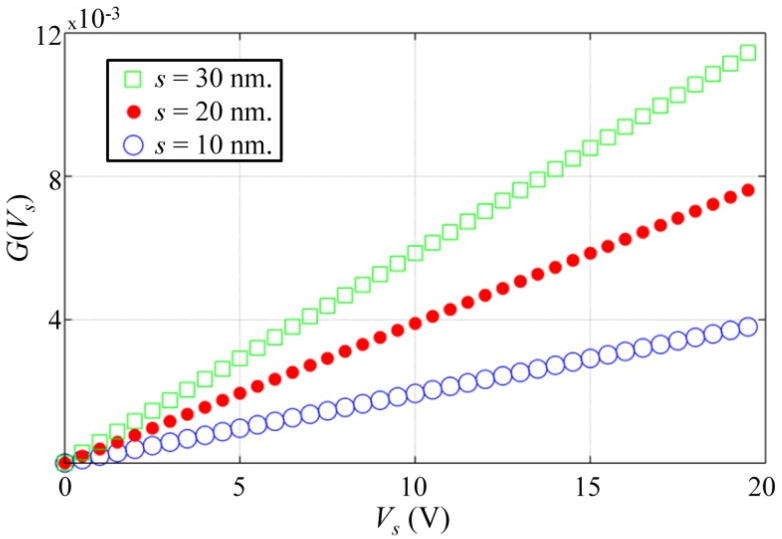
Simulation plot of *G*(*V*_s_) for multiple inter-grain separations (*s*) at *T* = 300 K and *d* = 0.02 cm. A definition for *G*(*V*_s_) is provided in Equation (2).

**Figure 2 polymers-12-01164-f002:**
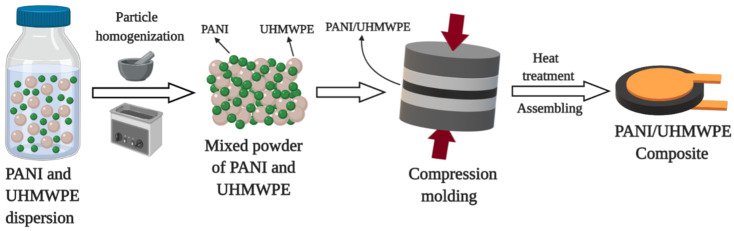
Sketch summarizing the preparation and assembly of the polyaniline/ultra-high molecular weight polyethylene (PANI/UHMWPE) composites.

**Figure 3 polymers-12-01164-f003:**
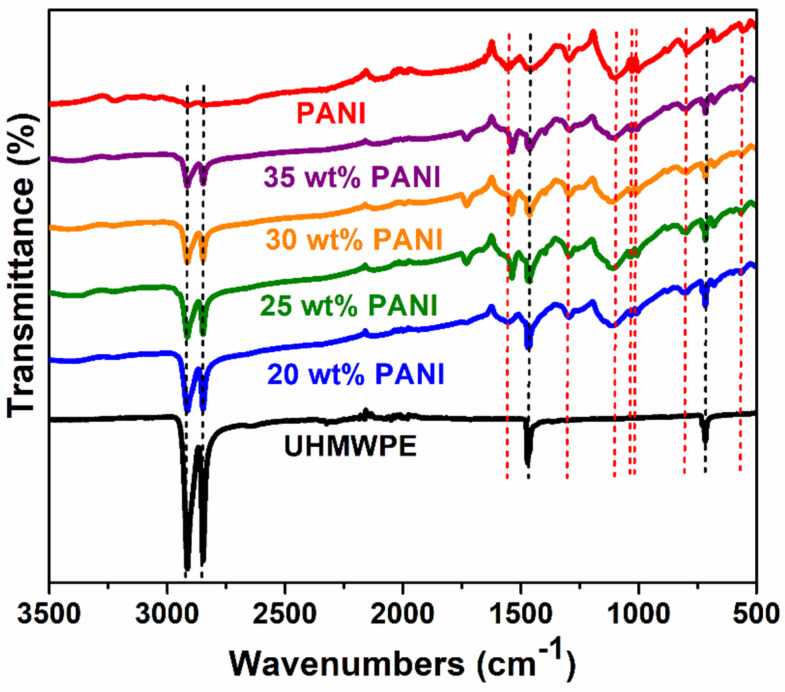
FTIR spectra of UHMWPE, PANI and PANI/UHMWPE composites.

**Figure 4 polymers-12-01164-f004:**
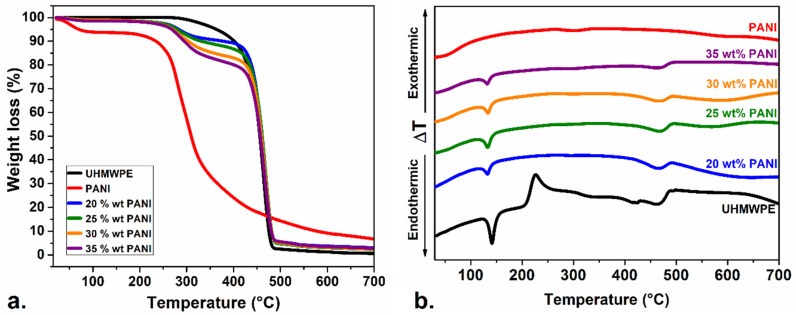
Thermogravimetric analysis (TGA) (**a**) and differential thermal analysis (DTA) (**b**) curves of UHMWPE, PANI and PANI/UHMWPE composites.

**Figure 5 polymers-12-01164-f005:**
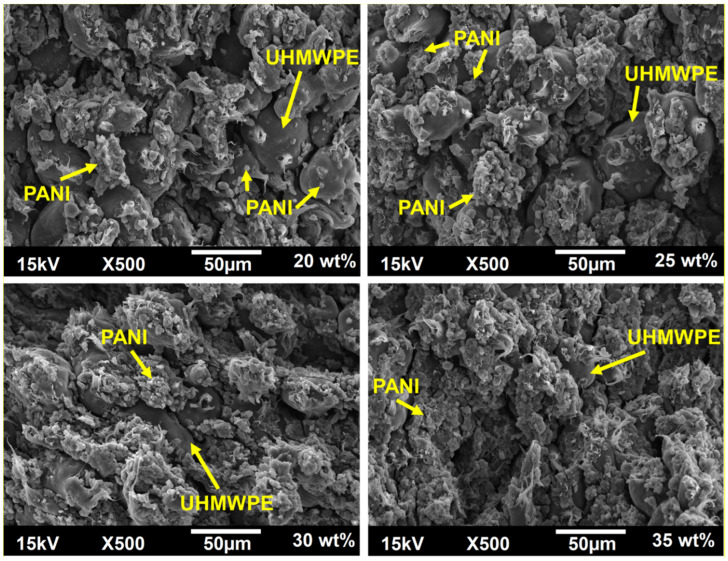
SEM images of composites with 20, 25, 30 and 35 wt % of PANI.

**Figure 6 polymers-12-01164-f006:**
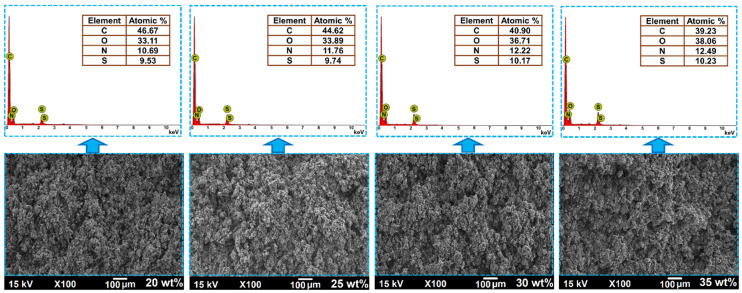
Energy-dispersive X-ray spectroscopy (EDS) microanalysis and SEM images of composites with 20, 25, 30 and 35 wt % of PANI.

**Figure 7 polymers-12-01164-f007:**
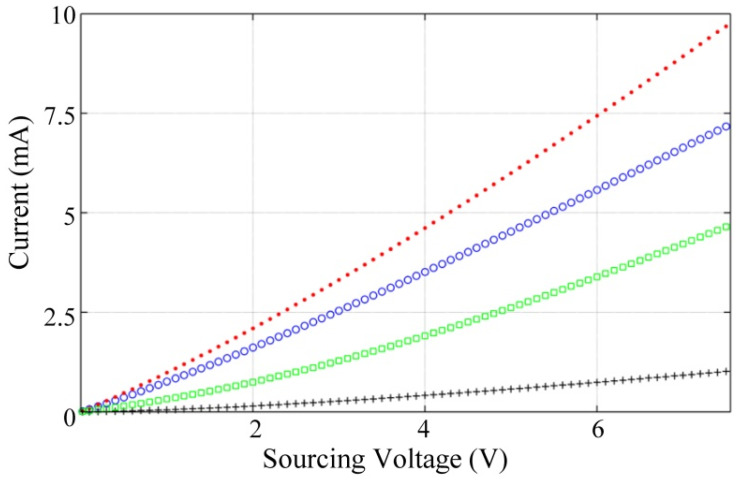
Current-voltage characteristic of the PANI/UHMWPE pellets at null mechanical stress with multiple weight content (wt %). Marker legend: 20 wt % (blue circles), 25 wt % (red dots), 30 wt % (black crosses) and 35 wt % (green squares).

**Figure 8 polymers-12-01164-f008:**
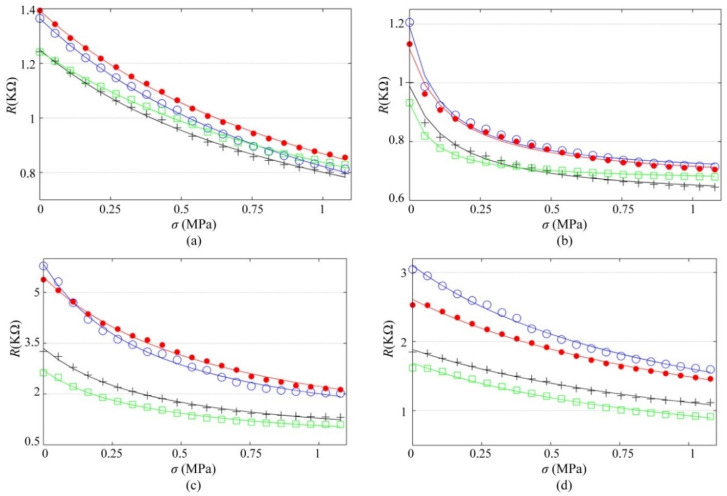
Piezoresistive response of the PANI/UHMWPE pellets at different test voltages, *V*_s_. (**a**) 20 wt %. (**b**) 25 wt %. (**c**) 30 wt %. (**d**) 35 wt %. Marker legend for *V*_s_: 1.25 V (blue circles), 2.5 V (red dots), 5 V (black crosses) and 7.5 V (green squares).

**Figure 9 polymers-12-01164-f009:**
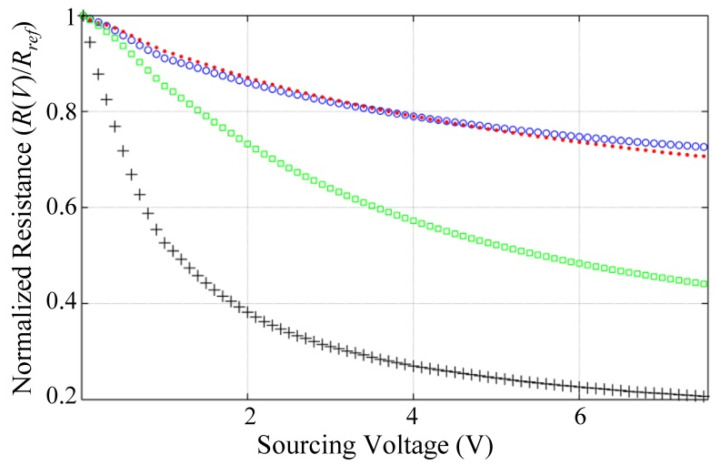
Normalized resistance of each PANI/UHMWPE pellet measured at null stress. Normalization obtained by dividing resistance data at each *V*_s_, *R*(*V*), by the resistance measured at *V*_s_ = 0.1 V, *R*_ref_. Marker legend: 20 wt % (blue circles), 25 wt % (red dots), 30 wt % (black crosses) and 35 wt % (green squares).

**Figure 10 polymers-12-01164-f010:**
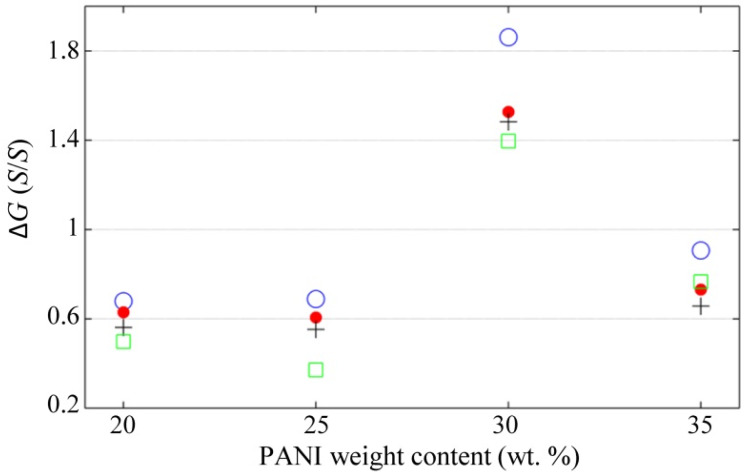
Conductance Variation, Δ*G*, in the PANI/UHMWPE pellets measured at different voltages. A definition of Δ*G* is available in Equation (4). Marker legend for *V*_s_: 1.25 V (blue circles), 2.5 V (red dots), 5 V (black crosses) and 7.5 V (green squares).

**Figure 11 polymers-12-01164-f011:**
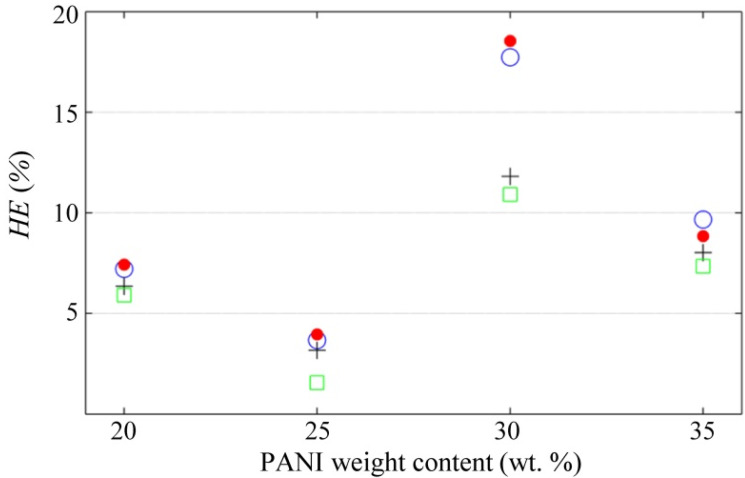
Hysteresis error, *HE*, of the PANI/UHMWPE pellets measured at different voltages. A definition for *HE* is available in Equation (5). Marker legend for *V*_s_: 1.25 V (blue circles), 2.5 V (red dots), 5 V (black crosses) and 7.5 V (green squares).

**Figure 12 polymers-12-01164-f012:**
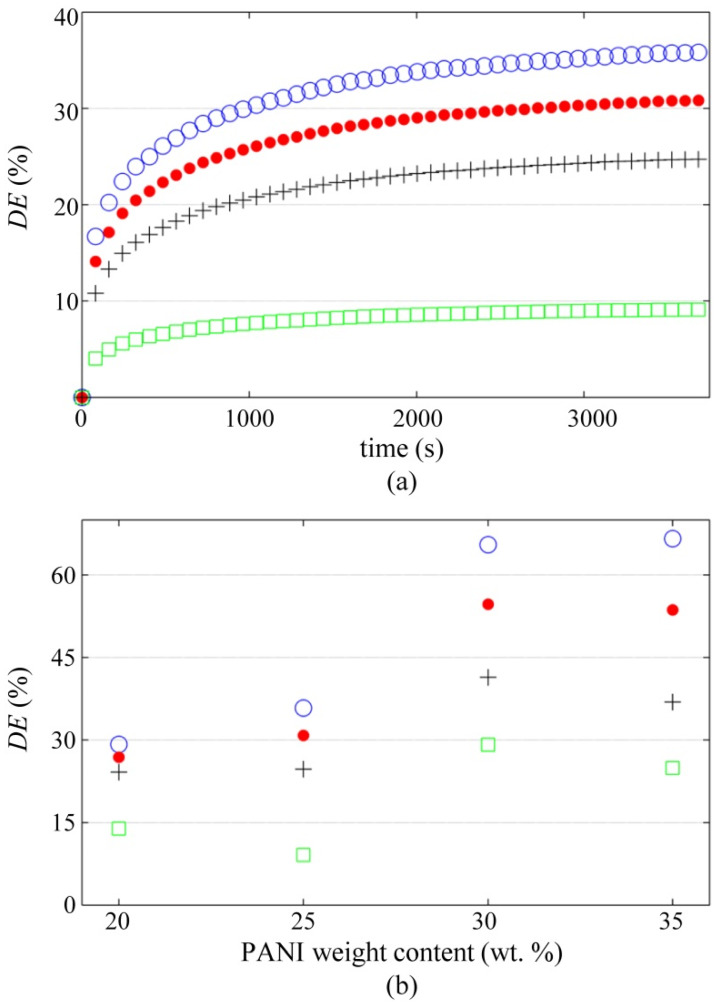
Drift error, *DE*, of the PANI/UHMWPE pellets measured at different voltages when loaded to 1.08 MPa. A definition for *DE* is available in Equation (6). (**a**) Drift error as a function of time for the 25 wt % specimen. (**b**) Drift error measured at *t* = 3600 s for all the PANI/UHMWPE pellets. Marker legend for *V*_s_: 1.25 V (blue circles), 2.5 V (red dots), 5 V (black crosses) and 7.5 V (green squares).

**Table 1 polymers-12-01164-t001:** Conductance and markers legend of PANI/UHMWPE pellets.

PANI/UHMWPE (wt %)	Conductance (*G*) at *V_s_* = 1 V and Null Stress
20/80	762 μS
25/75	986 μS
30/70	53 μS
35/65	320 μS

**Table 2 polymers-12-01164-t002:** Parameters resulting from fitting pellets resistance *R*(*σ*) to Equation (3). Individual fits were done at each voltage (*V*_s_).

PANI/UHMWPE (wt %)	*V*_s_ (V)	*R*_0_ (Ω)	*R*_1_ (GΩ·Pa)	*σ*_0_ (MPa)
20	1.25	186	1.38	1.17
	2.5	227	1.43	1.22
	5	316	1	1.08
	7.5	259	1.4	1.42
25	1.25	674	0.057	0.11
	2.5	655	0.069	0.15
	5	601	0.058	0.15
	7.5	663	0.022	0.08
30	1.25	470	2.15	0.4
	2.5	401	2.93	0.57
	5	422	1.22	0.41
	7.5	413	0.91	0.4
35	1.25	60	3.16	1.04
	2.5	52	3.28	1.28
	5	58	2.53	1.38
	7.5	63	1.83	1.13

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
