# Peer review of "Assessing the Influence of the Sourcing Voltage on Polyaniline Composites for Stress Sensing Applications"

_polymers, 2020, doi:10.3390/polym12051164_

Round 1

Reviewer 1 Report

The manuscript submitted by author related to synthesis, characterization and application of PANI composites in strain/stress sensing applications as an alternative to carbonaceous materials is a good effort made by author.

Few of the suggestion are listed below to improve the quality of script.

  1. I can see introduction section is too lengthy. Under the introduction section state the objectives of the work and provide an adequate background, avoiding a detailed literature survey or a summary of the results. Focus on a number of key references; do not overlook the earlier, seminal work.
  2. Under introduction section many sentences are without relevant references. Kindly correct it and insert these two relevant references in it. Journal of Science: Advanced Materials and Devices 1, 431-453, 2016; International Journal of Biological Macromolecules 89, 89-98, 2016.
  3. Provide the clear motivation of choosing Polyaniline and Ultra-high molecular weight polyethylene (UHMWPE) for the synthesis of PANI composites
  4. Why author have used this composite for strain/stress sensing applications?
  5. Why author used PANI? One of the main limitations of polyaniline is the loss of conductivity in neutral and high pH environment. Since polyaniline requires a large amount of protons attached to the polymer to be electrically conducting, it is a very poor conductor when the pH is greater than 5, which significantly limits its application. Justify?
  6. Provide the schematic diagram and explain the real chemistry behind the synthesis of PANI composite, types of reactions, bonding etc.
  7. What is optimum condition for study of piezoresistive model for PANI/UHMWPE composites?
  8. How will the ratio of PANI/UHMWPE will impacts the performance of piezoresistive response.
  9. In Figure 3. (b) TGA curves of UHMWPE, PANI and PANI/UHMWPE composites. I can think in case of composite its three step process. Just confirm it.
  10. In Figure 4. SEM images of composites with 20, 25, 30 and 35 wt.% of PANI.I cannot notice any significance change in SEM images. Compare the SEM images of composite with that of individual PANI and UHMWPE.

Author Response

Please see attachment, file: "response 1st reviewer.pdf"

Reviewer 2 Report

Comments :

The paper is dealing with applying PANI/UHMWPE polyblend as stress meter, basing on the resistance variation during pressing.  The contribution of resistance mainly came from the conducting PANI which is a protonic acid-doped polymer.  Therefore, its shape, distribution ( morphology ) in the UHMWPE matrix play important roles in deciding the sensitivity of the pressure meter.  However, the basic data of the purchased PAni, are not available, such as dopant type, which made the evaluation of the final product difficult.

  1. How did the authors know the weight ratios between 20 to 35 % of PANI in the polyblend are suitable to be a stress meter? ( page 1, line 24 ).
  2. In the introduction ( page 2, line 57 ), the authors mention CNT, a 1-D conductor and CB, point conductor, were applied in the same application, PANI is a point or 1-D conductor is dependent on the solvating capability of ethanol or its compatibility with UHMWPE. It is suggested that the authors can compare the results with the CNT and CB ( similar size and shape with PANI ) by mixing them with UHMWPE.
  3. PAni can demonstrate lots of types of morphologies based on the conditions of the electrochemical polymerization. Why did author choose the nanofibrous PAni in the studies? If PAni is in particle form or becomes a film on the nanofillers, the capacitance or other properties of the double-or triple layers could also change.
  4. Page 5, line 193. What is the acid-dopant of the commercial PANI? In accordance with IR spectrum demonstrated in Fig. 3(b), it could be some kind of aliphatic sulfonic acid.
  5. Page 5, line 194. MW 15.000 g mole-1 should be 15,000 g mole-1.
  6. Page 5, line 203. The solvation capability of ethanol on PANI and UHMWPE need to be confirmed separately, which would decide the morphologies of PANI in the polyblends.
  7. What is the melting point of UHMWPE, which can be found from DTA in the inset of Fig. 3(b) with wider range of temperature since melting can cause the destruction of the original morphology.
  8. 4-SEM. The PANI included in the blend seems to be fibrous. The SEM micrographs of neat PANI and UHMWPE need to provide to decide the distribution of PANI. The scale of the provided insets of Fig. 4 is too small.  Larger scale micrographs are necessary to examine the distribution.
  9. If EDX spectroscopic data are available along with SEM pictures, we can also understand the distribution of PANi by mapping the N-element.

Author Response

Please see attachment, file: "response 2nd reviewer.pdf"

Round 2

Reviewer 1 Report

The author have satisfactory revised the script.

Reviewer 2 Report

All questions were clearly answered.